# Risk of Death and Ischemic Stroke in Patients with Atrial Arrhythmia and Thrombus or Sludge in Left Atrial Appendage at One-Year Follow-Up

**DOI:** 10.3390/jcm11041128

**Published:** 2022-02-21

**Authors:** Katarzyna Kosmalska, Natasza Gilis-Malinowska, Malgorzata Rzyman, Ludmila Danilowicz-Szymanowicz, Marcin Fijalkowski

**Affiliations:** 1Cardiology Department of Saint Vincent de Paul Hospital in Gdynia, 81-348 Gdynia, Poland; katarzyn5@wp.pl (K.K.); katarzynakosmalska11@gmail.com (M.R.); 2First Department of Cardiology, Medical University of Gdansk, 80-210 Gdansk, Poland; n.gilis@gumed.edu.pl; 3Department of Cardiology and Electrotherapy, Medical University of Gdansk, 80-210 Gdansk, Poland; ludmila.danilowicz-szymanowicz@gumed.edu.pl

**Keywords:** atrial fibrillation, left atrial appendage thrombus, anticoagulation

## Abstract

Thrombus in the left atrial appendage is a contraindication for cardioversion. Sludge is considered similarly as threatening as thrombus; however, the risk of death and ischemic stroke in patients with atrial arrhythmia and thrombus or sludge is not well-known. This study focused on assessing the risk of death and ischemic stroke at one-year follow-up in patients with atrial arrhythmia and thrombus or sludge, as well as the effectiveness of anticoagulation in thrombus resolution. 77 out of 267 (29%) of patients who were scheduled for cardioversion were diagnosed with thrombus or sludge. The annual mortality in patients with thrombus or sludge was 23%. In the group without thrombus, the annual mortality was 1.6%. Overall, 17% of patients with thrombus or sludge experienced ischemic stroke. In patients without thrombus, the risk of stroke was 1%. Sludge increased risk of stroke compared to those without thrombus or sludge by 11% vs. 1%, respectively. No differences in mortality or stroke prevalence were observed between sludge and thrombus. Thrombus or sludge in the LAA have a poor prognosis. A diagnosis of sludge has a similar impact on risk of ischemic strokes as does a diagnosis of thrombus.

## 1. Introduction

Atrial fibrillation (AF) and atrial flutter (AFL) are the most frequent sustained cardiac arrhythmias, with a prevalence of about 2–4% in the general population; they have a 10% annual risk of death [1,2,3]. AF and AFL are associated with the development of left atrial appendage (LAA) thrombus, which is the main cause of stroke and systemic embolism [4,5,6,7]. In addition to thrombo-embolism, AF complicated by thrombus is characterized by a constellation of atherosclerotic risk factors that may predispose patients to serious clinical complications [8]. Several studies have stated that transesophageal echocardiography (TOE) is a conclusive imaging technique for excluding or confirming thrombus or dense spontaneous echo contrast in the LAA [9,10,11]. In subjects with atrial fibrillation, it is still unclear if dense spontaneous echo contrast (sludge) in the LAA is a robust independent risk factor for stroke, similar to thrombus formation [11,12,13]. Another recognized neurological complication is cognitive disorders: patients with thrombus or sludge in the LAA scored significantly lower on the Mini Mental State Examination (MMSE) than patients free from thrombus or sludge [14]. Anticoagulation is an established method for decreasing the risk of potentially fatal thromboembolism in AF/AFL [12,15,16]. However, many patients are still diagnosed with thrombus or dense spontaneous echocontrast in the LAA despite proper anticoagulation [14,17,18,19]. There are no specific recommendations, even in the latest guidance of 2020, on how to proceed with patients diagnosed with thrombus in the LAA. There are no recommendations to perform routine TOE control for assessing thrombus resolution, and the decision to perform TOE control is usually made by the physician and the patient when the AF/AFL is persistent before planned cardioversion [20].

This study aimed to answer the following relevant clinical questions: (1) How high is the risk of death and ischemic stroke in patients with diagnosed thrombus or sludge in the LAA at one-year follow-up? (2) Is a diagnosis of sludge as unfavorable as a diagnosis of solid thrombus? (3) Does thrombus resolution improve the patient’s prognosis?

## 2. Materials and Methods

From January 2016 to June 2017, 267 consecutive patients from two medical centers (St. Vincent Hospital in Gdynia and the First Department of Cardiology, Medical University of Gdansk) scheduled for direct current cardioversion (DCC) due to persistent AF/AFL were prospectively enrolled in the study. The exclusion criteria were arrhythmia shorter than 48 h, contraindications for TOE, and lack of consent. All of the included patients had undergone transthoracic and transesophageal echocardiography. The local ethics committee approved the study protocol, and written informed consent was obtained from all participants.

Ultrasound scanners (GE, Vivid S70, Horten, Norway) equipped with a 1.5–4.6 MHz transducer and a 3–8 MHz omniplane phase probe were used. In transesophageal echocardiography, precise scanning of the LAA was performed at the angles of 30°, 60°, 90°, and 120°; a transgastric two-chamber scan was performed at an angle of 90°, with careful adjustment of the gain and frequency in search of thrombus and dense spontaneous echocontrast (sludge) [20,21,22]. LAA early diastolic emptying velocity was also measured. As described in the literature, sludge was defined as an intracavitary echodensity with viscid gelatinous qualities that gave the impression of impending precipitation but without a discrete organized mass [23,24]. Echocardiography was performed and analyzed by two certified echocardiographers (researchers KK and MF) with appropriate intra- and inter-observer reproducibility. It is worth noting that our centers perform TOE before cardioversion in every patient, even with adequate anticoagulation. In the case of confirming both LAA thrombus and sludge, sinus rhythm reversal was postponed. Further therapeutic decisions depended on clinical conditions: 34 patients (44%) had chronic atrial fibrillation; they were prescribed standard oral anticoagulant therapy (OAC) and no TOE control. The remaining 43 patients (56%) underwent further attempts at cardioversion in the case of thrombus resolution with the use of three different methods of anticoagulation: vitamin-K antagonists (VKA) with rigorous control of INR, non-vitamin-K oral anticoagulants (NOAC) in doses recommended for AF/AFL, or therapeutic doses of low-molecular-weight heparin (LMWH) (enoxaparin 1 mg/kg bid). The latter group was subjected to control TOE three months later. The study followed up with the patients with and without thrombus or sludge for one year to evaluate deaths and thromboembolic events. The main follow-up data source was a telephone survey. In the case of patients who could not be contacted by phone, the patients’ general practitioners or the national healthcare system were queried. The local Ethics Committee approved the protocol of the study, and written informed consent was obtained from all participants.

Prospectively collected continuous data were analyzed and presented as a mean value and standard deviation (SD) or as a median and interquartile range (IQR). Categorical data were presented as a percentage. Normal distribution was verified by Kolmogorov–Smirnov test. Continuous data were compared by Student’s t-test or U-Mann–Whitney test depending on the distribution. Categorical data were compared by chi-squared test and Fisher’s exact test, as appropriate.

In order to find independent risk factors for death, a univariate logistic regression was performed, and then variables with *p* < 0.10 were included in the multivariate logistic regression. *p* value less than 0.05 was considered statistically significant. Data were analyzed using SPSS software v.21 (IBM, Chicago, IL, USA).

## 3. Results

In the entire group of 267 patients, 77 (29%) were diagnosed with thrombus or sludge. The baseline characteristics of patients with and without LAA thrombus or sludge are shown in Table 1. Among the group with thrombus or sludge in the LAA, 9% were not adequately treated with anticoagulants before TOE (either no anticoagulation, anticoagulation treatment shorter than three weeks, warfarin with TTR < 80%, or LMWH in prophylactic dose), 3% received therapeutic doses of LMWH, 54% received NOAC, and 34% received VKA.

### 3.1. Subsection

#### 3.1.1. Risk of Ischemic Stroke and Annual Mortality

After a one-year follow-up with the thrombus or sludge group, 18 patients had died (23%), and 13 (17%) had experienced an ischemic stroke. In the group without thrombus or sludge, three patients (1.6%) had died and two (1%) had experienced a nonfatal ischemic stroke (Table 2). The odds ratio (OR) for mortality in patients with thrombus or sludge was 17.7 (95% confidence interval [CI]: 5.0–62.3, *p* < 0.0001). The OR for ischemic stroke was 14.5 (95% CI: 3.1–67.7, *p* < 0.0001). The OR for both mortality and ischemic stroke was 14.8 (95% CI: 5.4–40.1, *p* < 0.0001). Kaplan–Meier curves are presented in Figure 1 and Figure 2 for mortality and ischemic stroke.

#### 3.1.2. Differences in Risk of Ischemic Stroke and Annual Mortality between Sludge vs. Appendage Free from Thrombus and Sludge

After a one-year follow-up with the group containing sludge patients, two patients had died (11%) and three (16%) had experienced an ischemic stroke. In the group without thrombus or sludge, three patients had died (1.6%) and two (1%) had experienced an ischemic stroke. The statistical differences are significant for stroke and borderline for one-year mortality—data are included in Table 2.

#### 3.1.3. No Significant Differences in Risk of Ischemic Stroke and Annual Mortality between Thrombus and Sludge

Patients with sludge in the LAA experienced fewer strokes (2 out of 19, 11%) than those with thrombus (9 out of 58, 16%). There were also fewer deaths in patients with sludge (3 out of 19, 16%) than in patients with thrombus (14 out of 58, 24%), but the differences were not statistically significant—data are included in Table 2. Baseline characteristics between patients with thrombus and sludge are also presented in Table 3.

#### 3.1.4. No Significant Differences in Risk Level between Successful and Unsuccessful Thrombus/Sludge Resolution

Patients with dissolved thrombus or sludge in the LAA experienced fewer ischemic strokes (2 out of 26; 7.7%) than those with persistent thrombus/sludge (2 out of 16; 12%). There were fewer deaths among patients with dissolved thrombus or sludge (4 out of 26; 15%) than among patients with persistent thrombus or sludge (4 out of 16; 25%), but the differences were not statistically significant—data are included in Table 2.

#### 3.1.5. No Significant Differences in Risk Level between Patients Treated with LMWH vs. OAC

Patients treated with LMWH had a lower number of strokes (1 out of 12; 9%) than those treated with OAC (12 out of 16; 75%). The number of deaths in patients treated with LMWH (2 out of 12; 19%) was also lower than the number of deaths in the OAC group (16 out of 63; 25%). Neither of these differences were statistically significant—data are included in Table 2.

#### 3.1.6. No Significant Differences in Risk Level between Patients Subjected to TOE Control vs. Those without Further TOE Control

After diagnosis of LAA thrombus, 34 patients (44%) were diagnosed with chronic arrhythmia and prescribed standard OAC. The TOE control was abandoned in this group. The remaining 43 patients (56%) underwent anticoagulation to dissolve thrombus. The latter group was subjected to further TOE control and was directed to undergo cardioversion in the case of thrombus resolution. Patients with thrombus or sludge had fewer strokes (4 out of 43; 9%) and deaths (7 out of 43; 16%) if they were further treated with thrombus resolution and cardioversion, but the difference was not significant. Patients who decided against further control and cardioversion had a higher rate of strokes (7 out of 34; 21%) and deaths (10 out of 34; 29%), but this difference was not significant either. Oral anticoagulants have high effectiveness in patients who have not previously been treated. This study suggests that implementing heparin for three months tends to be the most effective therapeutic option (continuing NOAC and VKA resulted in dissolving thrombus in 44% and 43% of patients, respectively, vs. 82% with LMWH treatment)—data are included in Figure 3.

## 4. Discussion

Administering transesophageal echocardiography before cardioversion revealed a significant complication: atrial thrombus in the LAA [3,14,25]. Despite the prevalence and potential devastation of atrial thrombus, there are no guidelines on how to treat these patients to decrease their risk of stroke and death. The problem of how to deal with a patient diagnosed with thrombus in the LAA is still unresolved. This uncertainty led the present study to conduct further observations of patients with AF/AFL who underwent TOE before planned cardioversion with respect to the main endpoints (strokes and deaths). It is worth mentioning that the annual mortality among patients from developed countries who have atrial fibrillation is approximately 10%, regardless of LAA thrombus diagnosis [1,2,3]. The present study results revealed that the risk of this outcome was 23% among patients with thrombus or sludge; this is compared to only 1.6% in patients with an LAA free from thrombus or sludge. The annual death rate (7.6%) in study group was slightly higher than in other studies, that could be explained by the relatively advanced age of our study group. This analysis revealed also a very high annual risk of ischemic stroke in thrombus and sludge patients: 17% vs. 1% in patients without thrombus or sludge. The annual risk of ischemic stroke in the general population of AF/AFL (for anticoagulated patients) is 1.4% for NOACs and 1.7% for VKA [1,2,3]. Thrombus in the LAA comes with a poor prognosis in terms of death and ischemic stroke. Therefore, the questions of how to treat these patients and whether the treatment improves the patient’s prognosis are of great clinical value. Oral anticoagulants have high effectiveness in patients who have not previously been treated. This study found that implementing heparin for three months was the most effective therapeutic option. A similar efficacy for OAC (41.5%) was revealed in a CLOT-AF study [26].

Another problem was that of classifying patients with dense spontaneous echocontrast (sludge). In this study, the presence of sludge significantly increased the risk of ischemic stroke; the number of deaths was also increased, but not significantly compared to LAA free from thrombus or sludge. Moreover, there was no statistically significant difference in the number of strokes or deaths between patients with a diagnosis of sludge versus a diagnosis of thrombus. Our results are similar to those published previously by Fatkin et al. and Berhardt et al. [27,28]. Paradoxically, these poor results yield easier clinical decisions, as differentiation between sludge and thrombus is often difficult [25,26]. Despite our expectations, dissolving LAA thrombus or sludge did not significantly decrease the number of ischemic strokes or deaths at one-year follow-up. Those observations would suggest that the risk of ischemic stroke or death in patients with atrial fibrillation is more complex, and abnormalities leading to thrombus formation, such as inflammatory mediators or endothelial dysfunction, regardless of the presence of thrombus, can worsen the patient’s prognosis.

The main limitation of this study is the relatively small number of enrolled patients. The sludge diagnosis is not also standardized and is subjective. Additionally, the method and duration of anticoagulant treatment of patients with a diagnosis of thrombus or sludge could be analyzed more thoroughly to answer an important clinical question: Is there a particular treatment of AF with thrombus that decreases the risk of death and stroke?

It is worth mentioning that both departments are participating in a new Polish multicenter prospective registry called Left Atrial Thrombus on TEE (LATEE). This study hopes to analyze the larger groups of patients contained in this registry, especially for the optimal management of LAA thrombus or sludge [29].

## 5. Conclusions

Diagnosis of thrombus or sludge in the left atrium in patients with atrial arrhythmias not only excludes the reversal of sinus rhythm but is also an important poor prognostic factor. A diagnosis of sludge in the LAA seems to have a similar impact on ischemic stroke and death prevalence at one-year follow-up as does a diagnosis of thrombus. Resolving thrombus or sludge yields fewer deaths and strokes in comparison to persistent thrombus or sludge, but the differences are not statistically significant.

## Figures and Tables

**Figure 1 jcm-11-01128-f001:**
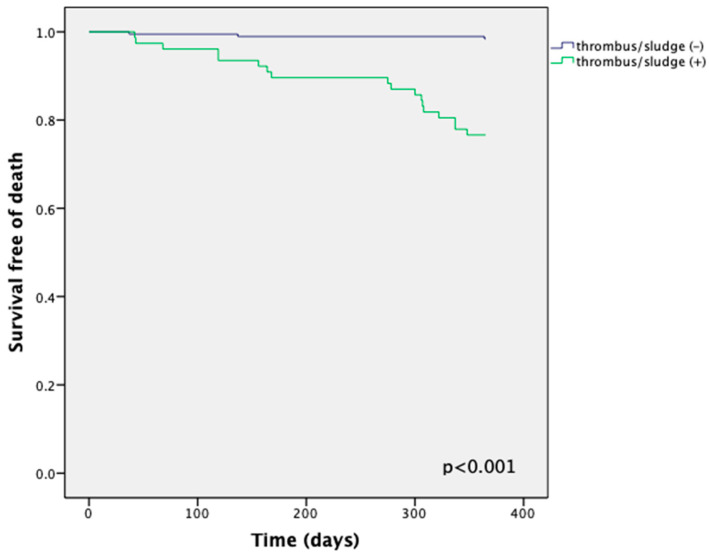
Kaplan–Meier curves for mortality of patients with and without LAA thrombus or sludge.

**Figure 2 jcm-11-01128-f002:**
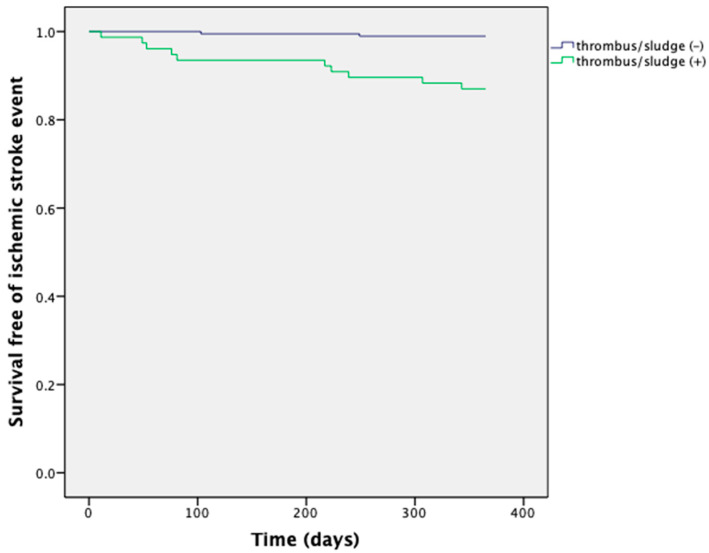
Kaplan–Meier curves for ischemic stroke of patients with and without LAA thrombus or sludge.

**Figure 3 jcm-11-01128-f003:**
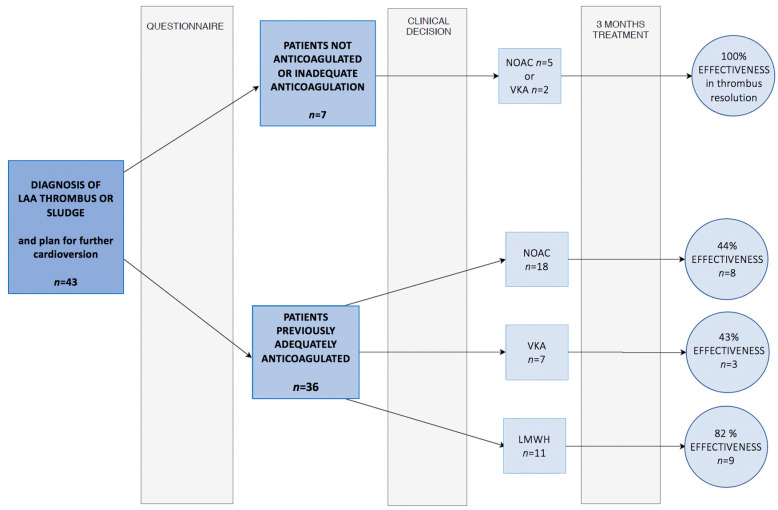
Graphic presentation of study decision tree, clinical course and treatment results. TOE—transesophageal echocardiography; LMWH—low-molecular-weight heparin; NOAC—non-vitamin-K antagonist oral anticoagulant; VKA—vitamin-K antagonist.

**Table 1 jcm-11-01128-t001:** Baseline characteristics of patients with and without thrombus or sludge in LAA.

	Thrombus or Sludge (+) *n* = 77	Thrombus or Sludge (−) *n* = 190	*p*
Clinical			
Age (years)	74.99 ± 8.17	73.01 ± 10.19	1.000
Female sex, *n* (%)	42 (54.54)	82 (43.16)	0.092
BMI (kg/m2)	29.06 ± 5.35	29.11 ± 4.94	0.942
CHA2DS2VASc	4.70 (4.00–6.00)	4.00 (3.00–5.00)	0.101
CHA2DS2VASc > 3, *n* (%)	61 (79.22)	138 (72.63)	0.264
First episode of AF/AFL, *n* (%)	23 (29.87)	92 (48.42)	0.006
Atrial flutter, *n* (%)	12 (15.58)	41 (21.58)	0.267
Arterial hypertension, *n* (%)	63 (81.82)	147 (77.37)	0.422
Diabetes, *n* (%)	35 (45.45)	68 (35.79)	0.142
Previous stroke/TIA, *n* (%)	17 (22.08)	42 (22.11)	0.996
Congestive heart failure, *n* (%)	39 (50.65)	72 (37.89)	0.056
Current smoking, *n* (%)	4 (5.19)	7 (3.68)	0.574
Echocardiography			
LVEF (%)	43.36 ± 12.48	46.24 ± 11.72	0.075
LA area (cm2)	30.18 ± 10.26	27.05 ± 5.22	1.000
LA spontaneous echocontrast, *n* (%)	56 (72.73)	62 (32.63)	<0.001
Low LAA velocities < 20 cm/s, *n* (%)	68 (88.31)	113 (59.47)	<0.0001
Aortic stenosis moderate-to-severe, *n* (%)	4 (5.19)	13 (6.84)	0.618
Aortic regurgitation moderate-to-severe, *n* (%)	2 (2.60)	6 (3.16)	0.808
Mitral stenosis moderate-to-severe, *n* (%)	3 (3.90)	1 (0.53)	0.040
Mitral regurgitation moderate-to-severe, *n* (%)	18 (23.38)	50 (26.32)	0.618
Laboratory			
Creatinine (mg/dL)	0.99 ± 0.38	1.01 ± 0.41	0.713
Hematocrit (%)	40.46 ± 4.54	40.63 ± 4.68	0.787
Platelets (G/L)	213.19 ± 54.87	220.07 ± 82.79	0.503
INR	1.90 ± 0.89	1.73 ± 0.72	1.000
aPTT ratio	1.34 ± 0.40	1.30 ± 0.33	1.000
Anticoagulation			
Anticoagulation, *n* (%)	70 (90.91)	168 (88.42)	0.555
NOAC, *n* (%)	37 (48.05)	122 (64.21)	0.015
Anticoagulation interruption in last three months, *n* (%)	12 (15.58)	11 (5.79)	0.010

TIA—transient ischemic attack; LVEF—left ventricular ejection fraction; LA—left atrium; LAA—left atrial appendage; INR—international normalized ratio; NOAC—non-vitamin-K antagonist oral anticoagulants.

**Table 2 jcm-11-01128-t002:** The comparison of mortality and ischemic strokes in different groups depends on thrombus or sludge diagnosis, thrombus or sludge resolution, and the treatment (Fisher’s exact test).

	Thrombus/Sludge (+) *n* = 77	Thrombus/Sludge (−) *n* = 190	*p*
Ischemic stroke event (%)	13 (17%)	2 (1%)	<0.001
One-year mortality (%)	18 (23%)	3 (1.6%)	<0.001
	Sludge (+) *n* = 19	No Thrombus or Sludge *n* = 190	*p*
Ischemic stroke event (%)	2 (11%)	2 (1%)	0.042
One-year mortality (%)	2 (11%)	3 (1.6%)	0.066
	Sludge *n* = 19	Thrombus *n* = 58	*p*
Ischemic stroke event (%)	2 (11%)	9 (16%)	0.722
One-year mortality (%)	3 (16%)	14 (24%)	0.539
	Thrombus resolution (+) *n* = 26	Thrombus resolution (**−**) *n* = 16	*p*
Ischemic stroke event (%)	2 (8%)	2 (12%)	0.628
One-year mortality (%)	4 (15%)	4 (25%)	0.454
	LMWH *n* = 11	OAC *n* = 31	*p*
Ischemic stroke event (%)	1 (9%)	4 (13%)	1.000
One-year mortality (%)	2 (18%)	5 (16%)	1.000
	TOE control (+) (Intension for thrombus resolution); *n* = 43	TOE control (**−**) *n* = 34	*p*
Ischemic stroke event (%)	4 (9%)	7 (21%)	0.198
One-year mortality (%)	7 (16%)	10 (29%)	0.181

**Table 3 jcm-11-01128-t003:** Baseline characteristics between patients with thrombus and sludge.

	Sludge *n* = 19	Thrombus *n* = 58	*p*
Clinical			
Age (years)	75.5 ± 9.3	74.8 ± 7.9	0.756
Female sex, *n* (%)	9 (47.4)	33 (56.9)	0.469
BMI (kg/m^2^)	27.7 ± 4.0	29.5 ± 5.7	0.197
CHA2DS2VASc	4.7 ± 1.4 5.0 (4.0–6.0)	4.7 ± 1.5 5.0 (4.0–6.0)	0.942
CHA2DS2VASc > 3, *n* (%)	16 (84.2)	45 (78.9)	0.618
First episode of AF/AFL, *n* (%)	6 (31.6)	17 (31.5)	0.994
Atrial flutter, *n* (%)	4 (21.1)	8 (14.3)	0.486
Arterial hypertension, *n* (%)	16 (84.2)	47 (82.5)	1.000
Diabetes, *n* (%)	9 (47.4)	26 (45.6)	0.894
Previous stroke/TIA, *n* (%)	3 (15.8)	14 (24.6)	0.537
Congestive heart failure, *n* (%)	10 (52.6)	29 (50.9)	0.895
Current smoking, *n* (%)	1 (5.3)	3 (5.3)	1.000
Echocardiography			
LVEF (%)	44.0 ± 11.5	43.1 ± 12.9	0.779
LA area (cm^2^)	29.1 ± 4.9	30.7 ± 12.0	0.620
LA presence of spontaneous echocontrast, *n* (%)	16 (84.2)	40 (70.2)	0.229
Low LAA velocities < 20 cm/s, *n* (%)	19 (100.0)	49 (86.0)	0.189
Aortic stenosis moderate-to-severe, *n* (%)	2 (10.5)	2 (3.4)	0.253
Aortic regurgitation moderate-to-severe, *n* (%)	1 (5.3)	1 (1.7)	0.435
Mitral stenosis moderate-to-severe, *n* (%)	0 (0.0)	3 (5.2)	0.571
Mitral regurgitation moderate-to-severe, *n* (%)	4 (21.1)	14 (24.1)	1.000
Laboratory			
Creatinine (mg/dL)	0.9 ± 0.3	1.0 ± 0.4	0.447
Hematocrit (%)	40.2 ± 4.4	40.5 ± 4.6	0.798
Platelets (G/L)	236.8 ± 62.4	205.0 ± 50.1	0.055
INR	1.8 ± 0.8	1.9 ± 0.9	0.645
aPTT ratio	1.3 ± 0.3	1.4 ± 0.4	0.603
Anticoagulation			
Anticoagulation, *n* (%)	19 (100.0)	52 (92.9)	0.567
NOAC, *n* (%)	10 (52.6)	27 (46.6)	0.645
Anticoagulation interruption in last three months, *n* (%)	5 (26.3)	7 (14.6)	0.299

## Data Availability

The data sets used and/or analyzed during the current study are available from the corresponding author on reasonable request.

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
