# Peer review of "Risk of Death and Ischemic Stroke in Patients with Atrial Arrhythmia and Thrombus or Sludge in Left Atrial Appendage at One-Year Follow-Up"

_jcm, 2022, doi:10.3390/jcm11041128_

Round 1

Reviewer 1 Report

The authors reported the results of a 267 patients who underwent TOE prior to planned cardioversion. They found left atrial appendage thrombus or sludge in 27% of the patients and it was associated with increased risk of stroke and mortality.

  • Was the study prospective or retrospective?
  • Line 73: Echocardiography was performed and analyzed by two 73 certified echocardiographers – Was each study performed by two physicians or the analysis was done by two? Were they blinded to each other interpretation?
  • How many patients the authors couldn’t reach by phone? What if general practitioner was not aware of outcomes?
  • Line 102: Only 9% were not adequate treated with anticoagulation. Why did the rest undergoing anticoagulation.
  • The rate of thrombus and/or sludge is extremely high in this population and significantly higher than what’s encountered in clinical practice. How do authors explain that?
  • 23% death rate at one year is extremely high. What were the causes of death?
  • What was the timing of stroke and death in both groups? It will be good to provide that at 30 days and 60 days

Author Response

Thank you for your efforts and time.

Marcin Fijalkowski

Reviewer 2 Report

Content suggestions:

  1. I would consider the examination of the effectiveness of anticoagulants by the method of anti-Xa or anti-IIa activity or INR in the case of taking VKA. Can the authors provide this information ?
  2. Do the authors have any information about concomitant medications with the influence on haemostasis (antiplatelet treatment…) ?

I think that the article would be a good inspiration for further studies.

The article could be edited after minor revision according to comments to the authors.

Author Response

(The authors gave the same response as above.)

Round 2

Reviewer 1 Report

The authors addressed most concerns.

Is the standard of care in Poland to perform TEE on all patients prior to cardioversion even if adequately anticoagualted?

The sentence (Heparin tended to 21 be more effective for thrombus resolution compared to oral anticoagulants) should be omitted. The study is not designed or powered to detect that.

In figure 3, what's the difference between the group of NOAC or VKA vs. NOAC alone or VKA alone?

Author Response

Thank the Reviewer for comments and effort made to review our manuscript.
